# Considering Risks to Researchers and Staff in Low-Resource Settings during Public Health Crises: A Proposed Conceptual Model

**DOI:** 10.3390/children10030463

**Published:** 2023-02-26

**Authors:** Krystle M. Perez, Muhammad Asim, Elliott M. Weiss, Gregory C. Valentine, Avinash Kavi, Manjunath S. Somannavar, Ibezimako Iwuh, Chikondi Chiweza, Kirkby D. Tickell, Benson O. Singa, Kristin Beima-Sofie, Maneesh Batra, Judd L. Walson, Rachel Umoren, Maureen Kelley, Sarah Saleem

**Affiliations:** 1Department of Pediatrics, University of Washington, Seattle, WA 98195, USA; 2Department of Community Health Sciences, Aga Khan University, Karachi 74800, Pakistan; 3Department of Obstetrics and Gynecology, Baylor College of Medicine, Houston, TX 77030, USA; 4KLE Academy of Higher Education and Research, Jawaharlal Nehru Medical College, Belagavi 590010, India; 5Texas Children’s Global Women’s Health Centre of Excellence, Lilongwe Private Bag B-397, Malawi; 6Department of Global Health, University of Washington, Seattle, WA 98195, USA; 7Kenya Medical Research Institute (KEMRI), Nairobi P.O. Box 54840-00200, Kenya; 8Center for Ethics in Health Care, Oregon Health and Science University, Portland, OR 97239, USA

**Keywords:** global health ethics, risks to researchers, public health crises

## Abstract

Human subjects research protections have historically focused on mitigating risk of harm and promoting benefits for research participants. In many low-resource settings (LRS), complex and often severe challenges in daily living, poverty, geopolitical uprisings, sociopolitical, economic, and climate crises increase the burdens of even minimal risk research. While there has been important work to explore the scope of ethical responsibilities of researchers and research teams to respond to these wider challenges and hidden burdens in global health research, less attention has been given to the ethical dilemmas and risk experienced by frontline researcher staff as they perform research-related activities in LRS. Risks such as job insecurity, moral distress, infection, or physical harm can be exacerbated during public health crises, as recently highlighted by the COVID-19 pandemic. We highlight the layers of risk research staff face in LRS and present a conceptual model to characterize drivers of this risk, with particular attention to public health crises. A framework by which funders, institutions, principal investigators, and/or research team leaders can systematically consider these additional layers of risk to researchers and frontline staff is an important and needed addition to routine research proposals and protocol review.

## 1. Introduction

Researchers and research staff are the face and heart of research in communities. They work at the interface of patients, communities, and health systems, yet the dominant focus within research ethics is on the protection of human subjects. While critical, this focus largely overlooks the ethical dilemmas, potential harms, and hidden burdens affecting researchers and frontline research staff.

Important work in the ethical conduct of global health research has carefully considered how the context of low-resource settings (LRS) can exacerbate the burdens of research for participants, even in ‘minimal risk’ research [1,2,3,4,5,6,7]. Participants face complex and often severe challenges in daily living due to poverty, geopolitical uprisings, sociopolitical, economic, and climate crises [7,8,9,10,11]. Researchers and frontline research staff in LRS experience similar risks as they work, and often live, within the same complex daily living environments and are exposed to the same underlying socioeconomic disparities [11,12,13,14,15]. However, there is a relative paucity of literature exploring risks to researchers and research staff as they complete research-related activities in these settings.

Risks are further compounded during public health crises, straining already overburdened healthcare systems and infrastructure throughout LRS. Risks such as exposure to infection, injury, staff shortages, and psychological harm and distress can be exacerbated during public health crises, as highlighted by the COVID-19 pandemic. Anticipating ongoing global public health crises, including pandemics, it is critical to appraise such risks to develop interventions to ensure equitable, ethical, and sustainable models by which essential research may continue when it is needed most.

Here, we propose a conceptual model by which to consider risks to researchers and research staff, inclusive of baseline experience of risk and public health crisis’ contribution to that risk. Given the lack of published data quantifying researcher risk during public health crises, we hope the model may drive further investigations into mechanisms to anticipate and modify researcher risks.

## 2. The Importance of Context in Assessing Specific Risks and Vulnerabilities in LRS

A growing body of international research ethics scholarship has underscored the importance of informing researchers’ ethical obligations and responsibilities with detailed knowledge and appreciation of the local context, communities, and health systems [2,4,5,6,9,16,17,18]. Guidance on risk assessment for research participants has been well-explored, weighing potential benefits of the research to the potential harms [19,20,21]. Appreciating the distinctive context of living conditions in LRS has helped inform thinking around the ancillary care obligations and risk-benefit assessment for research in LRS and, in turn, the ethical responsibilities of researchers and research teams working in such environments [4,11,12,13,14,15]. With this work, there is growing appreciation for the burdens that often fall to frontline research staff. These researchers often must navigate and respond to the complex needs of participants and families.

There is emerging, albeit limited, consideration of the potential risks and benefits of conducting research for researchers and research staff, especially those in LRS or during periods of public health crises [22]. For purposes of this discussion, a public health crisis is a situation associated with significant harm to community health and the economy, often with loss of life and affecting one or more geographical areas [23,24,25,26]. During such crises, local research teams have described numerous risks, including risk of physical harm (i.e., risk of illness while conducting research during an epidemic) and/or risk of emotional harm (i.e., psychological and/or moral distress at witnessing patients die due to lack of hospital resources in the midst of a global pandemic) [22]. Additional ethical considerations relate to the complex relationships frontline researchers navigate when conducting research-related activities within their own communities [16,22,27]. It is also important not to assume that risk is limited to one type of research—such as the psychological and emotional risk in conducting sensitive, qualitative research—but is prevalent throughout a range of study designs including quantitative, clinical, observational, and translational research.

Environments in which researchers and research staff live and work are critical to understanding baseline experienced risks. Such baseline factors are highlighted in Figure 1 and include: (1) political, cultural, and physical environment; (2) institutional and working environment; (3) community and local support networks; and (4) domestic environment and living conditions. Political, cultural, and physical environments provide the context from which subsequent baseline factors of risk are derived. For example, institutional local policies informing a research staff’s work environment may reflect political (in)stability, cultural norms, and/or physical environments; weaker community support networks within environments of political oppression, cultural division and/or physical isolation; and impoverished domestic environments with limited access to basic resources. Importantly, many of these baseline factors are not modifiable, especially in the short-term.

Despite increasing awareness of such risks and ethical dilemmas to research teams in LRS, we lack models for evaluating, anticipating, and mitigating these risks. While risks to researchers and frontline staff are most obvious in LRS, ethical analysis of risks has applicability well beyond LRS. For example, while the COVID-19 pandemic highlighted daily, pervasive ethical dilemmas and risks faced by research teams in LRS, it also served as a substantial threat to public health and societal well-being in high-resource settings that resulted in enough researcher team risk that as much as 80% of the pre-pandemic human subjects research was halted [28]. Literature has also highlighted ethical dilemmas of frontline staff in higher income settings in the “age of austerity” when faced with dilemmas common in LRS, such as socioeconomic instability and reduced human resources to deliver quality care [29,30]. Thus, reducing risks across all research settings is paramount, with research teams working within LRS most susceptible to risk given political and socioeconomic environmental contexts.

## 3. Identifying Frontline Staff Experiences and Assessment of Potential Risks

While awareness of the challenges and ethical dilemmas experienced by research staff is increasing [16,17,18,27,31,32], there is a lack of robust assessment of the full range of risks and drivers of risk for research teams. Attempts to characterize these as occupational risk is a useful starting point, but limited in scope [25,33]. Classification as occupational risk may not fully acknowledge the moral distress and ethical dilemmas faced by researchers and research staff.

Researcher and research staff risks in LRS cannot be fully distinguished from background environmental risks of daily living in contexts of political crises, conflict, or poverty. Risks may be underreported due to political tensions, institutional policies, culturally accepted power imbalances between staff and study leadership, and/or or job insecurity. Dedicated research teams often feel the pressure of study timelines, perhaps taking risks to meet recruitment goals. Staff or early career researchers on precarious contracts may fear disappointing study managers and losing employment. These power imbalances, both between and within researcher teams, have been well described in the literature as a significant contributor to researcher risks [22,34]. These power imbalances have also likely contributed to the relative paucity of data on the topic.

Frontline research staff also experience risks to their mental well-being [22]. Notably, it can be particularly difficult to account for the psychological harms of working in research in LRS as these harms are often hidden and cumulative over time. A recent research ethics study, embedded within research activities within the Childhood Acute Illness and Nutrition (CHAIN) Network [35], followed research teams prospectively to better understand the ethical challenges and experiences of frontline researchers and staff. Frontline researchers were often from local communities, spent substantial time getting to know participants and their families, and reported a strong sense of obligation to help respond to complex needs and health problems, even well beyond the study focus. Many struggled with the scope and complexity of needs they witnessed and often felt powerless to help in a sustainable way. Feeling a strong sense of responsibility but lacking the power and resources to respond adequately led to significant psychological and moral distress [18]. A recent systematic review by Steinhert and colleagues confirmed this complex moral distress experienced by frontline research teams in LRS, arising from feelings of guilt and role conflicts when participants asked for help [22].

Notably, while researchers and research staff may experience similar dilemmas and risk of harm compared to frontline clinical staff, additional considerations should be given to differences in the non-clinical research staff experience of moral obligation, disparate access to resources, and additional susceptibility to power imbalances resulting in fears and hesitancy to say no or voice concerns. While healthcare professionals and staff may experience some degree of overarching support, via national or international clinically based organizations for example, non-clinical researcher team cadres may not benefit from such organizational support. Additionally, risk of harm in being clinically deployed amidst public health crises is weighed against the direct benefits to the patients and communities for whom you directly care. However, the benefits of conducting research in public health crises—ranging from development of a new intervention or to improve understanding of a condition—are (1) less immediate, (2) highly variable between studies and methods, and (3) may sometimes never come to fruition. Thus, the balance of risks and benefits of working within LRS and during a public health crisis, such as a pandemic, are distinct for clinical versus non-clinical, research staff. There are significantly less immediate benefits experienced by a non-clinical research team member as compared to their clinical counterpart.

## 4. Exploration of Risks to Researchers and Staff in LRS during Public Health Crises

Assessment of researcher team risk must also account for ongoing or emerging public health crises within the larger context. In addition to fueling job insecurity, power differentials, and moral distress, researcher risk may include sexual harassment, risk of physical harm, and/or political repression for participating within research activities within a conflict, post-conflict, or oppressive political environment [22]. Additional risk of physical harm, contamination, illness and/or death may be experienced during epidemics/pandemics or other disaster events.

Below, we highlight examples of risks to research teams in LRS conducting important research within three main public health crisis categories as depicted in Figure 1: (a) epidemics and pandemics, (b) geopolitical uprisings and conflict, and (c) human and natural disasters. Each of these serves to add layers of risk to existing baseline contextual risks. Specifically, public health crises threaten public health, contribute to socioeconomic instability; propagate geopolitical instability; and/or perpetuate health, financial, and power imbalances. Collectively, these additional layers serve to increase the risk experienced by the local researcher teams.

### 4.1. Epidemics and Pandemics

Perspectives of local research teams in LRS are minimally represented in the literature. However, categories of distress experienced by healthcare providers in high-resource settings during an epidemic have been described and include fear of infection, societal stigma and isolation, and dissolution of trust in and from their organization [36]. While Ebola outbreaks have been recurring for decades, the 2014 Sierra Leon Ebola epidemic resulted in a delayed, though ultimately rapid increase in research efforts triggered by the ongoing spread of Ebola to high-income countries and further fears of spread [37]. A study of health managers and staff working within Ebola treatment centers in Sierra Leon described their experiences in working to protect their homes and communities, as well as strategies that were helpful including supportive supervision, peer support, improved communication and systems to maintain and rebuild trust within the community [36]. Researchers and research staff may experience additional layers of ethical dilemmas related to potential lack of training and/or career opportunities, and a better understanding and evaluation of these risks of harm and the perceptions of them among research teams in LRS are needed.

Similar experiences and perceptions have been described among healthcare staff and researchers surrounding past epidemics of H1N1 [38], Zika virus [39], and SARS [40]. Thus, while context-specific risks and ethical considerations are likely, there are also shared experiences across settings within the context of an epidemic. Such themes are likely shared among researchers and research staff facing a global pandemic such as COVID-19. In LRS, researchers and frontline staff risks were magnified and multidimensional during the peak of the COVID-19 pandemic with layers of individual risk, societal stigma, and psychological distress. For example, the lack of access to personal protective equipment (PPE) despite exposure risk, or, alternatively, funding of PPE for researchers and research staff while their clinical colleagues had none were common ethical conundrums that paralleled subsequent, disparate COVID-19 vaccination access between healthcare providers and research team staff. Furthermore, efforts by researchers in high-resource countries to consider these risks in LRS, by halting research efforts or mandating protections in these LRS for example, also resulted in discrepant policies that served to underscore ongoing colonialization in global health research, especially when local culture and policies were poorly aligned with high-resource country recommendations. At an individual level, economic consequences of being unable to work (in research) had critical repercussions on researchers and frontline research staff.

### 4.2. Geopolitical Uprisings and Conflict Zones

The intersection of conflict and warfare blocking ongoing research activities, including early HIV research efforts [41], are known risks among the global community. Warfare and political riots, for example, not only impair local infrastructure and security services, but also the safety of local research staff and healthcare providers. The activities of research staff—e.g., traveling through unstable areas for data or sample collection—may expose them to risks, such as car jackings; being caught in gunfire; or being detained by military, paramilitary, or police. As previously mentioned, staff may feel reluctant to raise concerns given job insecurity and many local researchers may be motivated by strong commitments to the community. While studies are often suspended during severe political crises, others continue despite the danger, given the importance of the research for addressing a public health emergency. Examples cited from this group of authors’ experiences include critical research on tuberculosis and malaria prevention and COVID-19 infection surveillance continuing along the Thailand–Myanmar border despite the war in Myanmar or background political tensions. These underlying contextual factors contributed to distrust between the community and research teams, with participants becoming aggressive towards research staff when approached for consent. In our experience, these dangers must be weighed against the value of the research for improving health for those in the region and beyond. It is essential to consider the risks to particularly vulnerable team members who may not feel free to decline engaging in research activities. Nonetheless, local research team involvement is arguably the only hope for successful research efforts during times of political instability; they offer important context, often with pre-existing relationships and trust built with local communities [42].

### 4.3. Man-Made and Natural Disasters

Psychological distress and fears among first responders to terrorism, radio nuclear disasters, earthquakes, and hurricanes have been well documented, albeit in higher resource settings [43,44,45,46]. Mitigating factors, also explored in high-resource settings, include the facilitation of communication channels and peer and mentor support networks, similar to mechanisms described for epidemic categories of research team risk. Nonetheless, despite the seemingly increasing threat of human and/or natural disasters, there is limited data on the experiences and ethical dilemmas faced by researchers and research staff working in LRS after such a disaster. A recent article called for improved guidance given insufficient Institutional Review Boards algorithms to consider disaster-related research protection of communities, both as it pertains to participants and research teams [47]. Despite the lack of clear guidance, further understanding the implications of a disaster on a community—such as impacts on food, disease, water, carcinogens, or other human health factors—are needed.

## 5. The Conceptual Model of Risks to Researchers and Research Staff

Drawing on work evaluating social determinants of health [48], research team risk begins with consideration of their lived environments in the absence of research activities—as described above. Such baseline risk factors (depicted in the yellow vertical boxes) are critical to understanding context for potential downstream researcher team risk and are represented in the model as the individual threads to a larger rope. The more challenged an environment, the thinner the thread(s) and, thus, the thinner the rope. The strength of the rope metaphorically represents the relative buffer against additional strains endured by a research team. When baseline environments are strained, dynamic, and/or uncertain, additional public health threats serve to increase risk to researchers and research staff. Public health crises (depicted via the horizontal row of green boxes) will contribute universally to risk by threatening public health, socioeconomic stability, geopolitical stability and/or contribute to power imbalances. Collectively, these threats (depicted in the horizonal blue boxes) are represented as scissors that threaten to cut the metaphorical rope. If already thin, even a relatively small cut to the rope could have devastating implications regarding risk.

Importantly, research efforts should focus not only on quantifying researcher team risks in these settings, but also highlight mechanisms to protect research staff from these risks. Example mechanisms by which to prepare for and respond to additional risk are highlighted in the orange box and are represented by a cylindrical sheath protecting the rope from being cut. While baseline risk may be relatively predictable within certain settings, the additional threat of public health crises may or may not be. Ultimately, the strength and resilience of the rope—derived from both baseline and protective factors—represent the risk of harm to the research team. We acknowledge that protective factors listed in the model are only examples with further investigation into effective risk modifiers urgently needed.

As the categories of risks were developed by review of the limited literature and via the shared experiences of the authors, we acknowledge this model is not all-encompassing. Furthermore, exploration of feasibility and implementation are necessary. Nonetheless, we hope the model may serve to propel ongoing research in this field forward with improvements anticipated with more in-depth research on this topic.

## 6. Who Is Responsible for Implementation of the Model and Assessing Emerging Researcher Team Risk and Risk Modifiers?

Historic and recent examples of public health crises highlight the need for specific ethical considerations for researchers and research staff working in LRS. However, who it is that is responsible for accounting for such risks beyond the narrower occupational risks remains ill-defined. Institutional or ethics review boards or committees have not traditionally taken responsibility or governed researcher team risks and have been ill-equipped to assess and address emerging risks [22]. Additionally, while lead investigators are expected to consider the wider ranges of risks study teams might be exposed to during a study, oversight is lacking with potential conflicts of interest between a lead investigator and research team. Ultimately, shared responsibility between institutions, grantors, principal investigators, or leads of local research teams may be most appropriate with significant input from local research leadership. Ethical review committees are well-suited to oversee such shared responsibilities given existing infrastructure and experience with respect to protection of human subjects.

Responsible parties need be aware of not only potential categories of risk as proposed in our model, but also potential mechanisms to mitigate or prevent them. Ongoing systematic investigation into contributors of risk are critical to identify ideal prevention and mitigation strategies. For example, existing literature highlighting power imbalances as contributors to risk might be offset by mandating a system of checks-and-balances of power among the research team(s). Additional mitigation strategies may include assurance of communication platforms for all research team members and establishment of safe channels of communication, where concerns may be (anonymously) reported and reviewed. Safe and open lines of communication, even amidst uncertain sociopolitical contexts, may also bolster trust within and between research teams. Some of these strategies might be more important when work is being done within settings where distrust is higher and/or mechanisms of communication are poor, such as post-conflict or autocratic study contexts. Thus, systematic appraisal of the potential contributors and modifiers of risk to research teams is of vital importance to establish evidence-based mitigation or preventive strategies. Appraisals and recommendations should be context-specific, and local team experiences should be highlighted.

## 7. Conclusions

The ongoing COVID-19 pandemic, climate change, and recurring geopolitical crises have highlighted the need to explore the experiences of research teams working within LRS and beyond. Systematic ethical analyses of dilemmas and risks faced by researchers and research staff engaged in implementing research activities during public health crises are limited. Such an ethical analysis should be a focus of ongoing research, especially for researchers and research staff working within LRS. Investigations on staff risk will also need to account for environmental influencers on researcher staff decisions. Emerging work on global health decolonialization and strategies for building trusting, equal research partnerships between the Global North and South will be integral to begin to shift such power differentials [49].

Our proposed conceptual model may serve as one mechanism to categorize types and explore predictors of risk—both unique and shared between settings and research teams. The proposed model acknowledges not all contributors to research team risk, such as baseline environments, are immediately modifiable. However, additional layers of risk—such as from ongoing or emerging public health crises—may be anticipated and, therefore, mitigated. More detailed exploration of these risk factors and risk modifiers may facilitate development and implementation of a toolkit to minimize risk to researchers and research staff, especially during public health crises.

Ethical review committees may be well-suited to oversee risk-reduction strategies for research teams. Based on previously described research team experiences, potential mitigation strategies to explore include mechanisms to improve communication, supportive supervision, peer support networks and routine platforms by which trust can be established and/or rebuilt within research teams and the community at large. Exploration of risks and ethical dilemmas faced by researchers and research staff in LRS may serve to improve the sustainability and successful implementation of research endeavors globally. With ongoing public health crises anticipated, there is urgency in protecting researcher staff working within LRS where the “rope” is often thin at baseline.

## Figures and Tables

**Figure 1 children-10-00463-f001:**
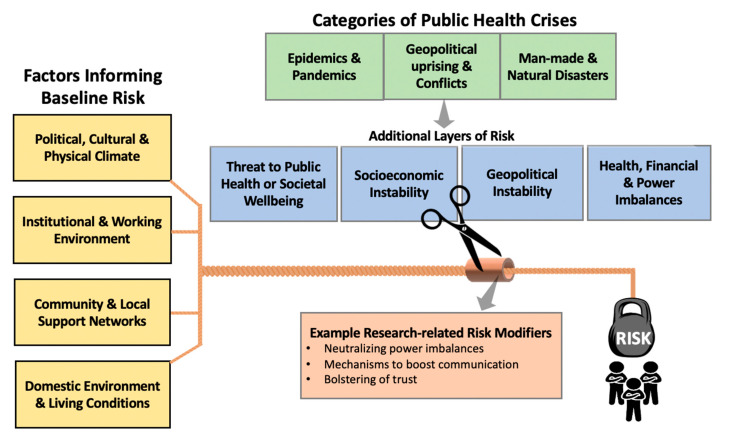
Conceptual model of contributors and potential modifiers to researchers and research staff risk. Beginning from the left of the figure, the yellow boxes are categories of baseline risk that are represented as individual threads that form a rope. Various threats (and protections) to the rope are encountered that translate to risk for a research team. “Risk” in this model is depicted as a weight suspended over the research team by the metaphorical rope. Beyond the strength of the rope derived from the thickness or thinness of its individual threads, the integrity of the rope may be compromised by “cuts” occurring in the setting of public health crises depicted in the green horizontal boxes, which lead to additional layers of risk described within the row of blue boxes and are represented as scissors. Just as these additional layers of risk threaten to cut the rope, protective factors—represented by a sheath—safeguard the rope from being cut. Examples of such risk modifiers are listed within the orange box.

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
