# Peer review of "Considering Risks to Researchers and Staff in Low-Resource Settings during Public Health Crises: A Proposed Conceptual Model"

_children, 2023, doi:10.3390/children10030463_

Round 1
Reviewer 1 Report
This study is supported by the most recent and well-represented literature and provides a good review of the literature on the topic, also in the context of on-going public health crisis such COVID-19.
However, the study does not provide an deep analysis, probably because the literature on the topic is often scarce. The model proposed could represent a valuable research tool, but it needs other data and perhaps adaptation of existing tools to be used; as the authors themselves said, it is a simplified model that needs a more detailed investigation on its feasibility. The review also does not give data or figures on the theme but it seems to be a way to raise awareness on an emerging problem. The conclusion and proposed model is not satisfactory. I think the review could be improved and enriched with data and figures available, to make sure the reader understands the relevance of this topic and the magnitude of the problem. It is though a starting point for further research in this area of global health.
Author Response
Response to Reviewer 1:
- We appreciate and agree with reviewer 1 that further in-depth analyses and data are limited. We have tried to further underscore the paucity of literature on the topic throughout the manuscript.
- We also believe the model proposed could be a valuable research tool, and we are hopeful data and adaptation of existing tools can occur following further investigation.
- We confirm that an objective of the manuscript is to highlight the problem, and we have inserted language to highlight the purpose of the model and its limitations, acknowledging the minimal data available.
- We have inserted a section to better describe the model, its relevance, and the limitations.
Reviewer 2 Report
This paper proposes an innovative model for identifying and reviewing the risks to researchers working in low resource settings, and in the particular circumstances of epidemics and pandemics, geopolitical uprising and conflicts, and man-made and human disasters.
This is a potentially valuable contribution to the research ethics literature. The model is well thought out.
However, at present the paper does not focus on the model, which is presented as a stand-alone figure with a very long caption. Instead, the paper claims to be a literature review, although there is no description of the methods used to search and review the literature and it does not meet the usual criteria for a systematic, scoping or narrative literature review.
My suggestions for strengthening the paper are to:
· Change the title to reflect the content with a focus on the model for assessing risk to researchers in low resource settings
· Put much of the current material into a Background section that:
o Explains the risks to researchers in low resource settings
o Explains how these are exacerbated by public health crises, thereby justifying the presence of these in the model
· Describe the model in detail
· Discuss further how it might be implemented
Corrections are needed regarding minor grammatical errors throughout the paper and in places, empirical claims are not referenced.
Author Response
Response to Reviewer 2:
- We appreciate reviewer 2’s input that the model is well thought out, and we have inserted a subsection dedicated to describing the model, the relevance, and its limitations in the manuscript.
- We agree this was not a systematic, scoping narrative review, and, thus, there is no methodology portion. We did change the title as suggested.
- To avoid a lengthy background section, we reformatted subheadings and bolstered content to better highlight (a) baseline risks to researchers in low resource settings, (b) how these risks are exacerbated by public health crises, and (c) the conceptual model, its relevance, and limitations
- We reviewed again to correct the minor grammatical errors.
- We explicitly stated when claims were supported by the shared experience of the author group, noting the limitations in published data on the topic.
Reviewer 3 Report
Based on a literature review, the authors highlighted the importance of risks to researchers and staff in low resource settings during public health crises and presented conceptual model to characterize the drivers of these risks. They started from the premise that contextual characteristics of low-resource settings (LRS) could exacerbate the burdens of research for participants, even in ‘minimal risk’ research. The text was structured in eight topics. I suggest to distribute the content in five topics: (1) The importance of context in assessing the specific risks and vulnerabilities in LRS; (2) Identifying frontline researchers’ experiences and assessment of potential risks; (3) Exploration of risks to researchers and staff in LRS during public health crises; (4) Who is responsible to assess researcher risk, especially as public health crises unfold, and what are possible modifiers?; (5) Conclusions. In the topic (3), I suggest to include the content referred to (3.1) Epidemics and pandemics; (3.2) Geopolitical uprisings and conflict zones; and (3.3) Man-made and natural disasters. As these three points represent subtopics of the topic 3, I am afraid that this structure is more useful and friendly for the Children’s readers.
The text is well written and the content is relevant for all researchers interested by dilemmas and risks faced by researchers and research staff. In the end, the authors presented a conceptual model that might be useful for examining the relationships among different risk predictors and categories.
Author Response
Response to Reviewer 3:
- We greatly appreciate Reviewer 3’s appreciation of the topic and its relevance.
- We restructured the subheadings to be clearer for Children’s readers per our understanding of reviewer 3’s suggestions.